# Development and Analysis of Silver Nitroprusside Nanoparticle-Incorporated Sodium Alginate Films for Banana Browning Prevention

**DOI:** 10.3390/nano14030292

**Published:** 2024-01-31

**Authors:** Lina Zhang, Anbazhagan Sathiyaseelan, Xin Zhang, Yuting Lu, Myeong-Hyeon Wang

**Affiliations:** 1Department of Bio-Health Convergence, Kangwon National University, Chuncheon 24341, Republic of Korea; 1911880536z@gmail.com (L.Z.); sathiyaseelan.bio@gmail.com (A.S.); zhangxin199708@gmail.com (X.Z.); yutinglu0614@gmail.com (Y.L.); 2KIIT (Kangwon Institute of Inclusive Technology), Kangwon National University, Chuncheon 24341, Republic of Korea

**Keywords:** silver nitroprusside nanoparticles, sodium alginate films, banana, shelf life, antibacterial activity

## Abstract

Banana (*Musa acuminate*) has been popular among consumers worldwide due to its rich nutrients and minerals. However, bananas are highly susceptible to the physical and biological factors that lead to postharvest loss during transportation and storage. In this work, novel sodium alginate (SA) films incorporated with silver nitroprusside nanoparticles (AgNNPs) were prepared to extend the shelf life of bananas through antibacterial and antioxidant coating. The results exhibited that AgNNPs were cubical and that their size was <500 nm, with metal composition being Ag and Fe. Additionally, the incorporation of AgNNPs in the SA film was seen in FE-SEM and zeta analysis, with an average size of about 365.6 nm. Furthermore, the functional and crystalline properties of AgNNPs were assessed through FTIR and XRD. Transmittance testing of the SA-AgNNPs films confirmed they have good UV barrier properties. SA-AgNNPs films exhibited excellent high antibacterial activity against foodborne pathogens including *L. monocytogenes*, *S. enterica*, and *E. coli* at the concentration of 500 µg/mL. Moreover, during the storage of bananas, SA-AgNNPs nanocomposite coatings act as a barrier to microbial contamination and slow down the ripening of bananas. As a result, compared with SA-coated and uncoated bananas, SA-AgNNPs-coated bananas exhibited the lowest weight loss and lowest total bacterial colonies, thus greatly extending their shelf life. Particularly when coated with SA-AgNNPs films, total bacterial colonies (TBC) in the banana peel and pulp were as low as 1.13 × 10^3^ and 51 CUF/g on the ninth day of storage, respectively. Our work offers an efficient strategy to improve the quality of bananas during the postharvest period, with extensive applications in fruit preservation and food packing.

## 1. Introduction

Food coatings are usually achieved through packaging or spraying onto the surface of foodstuffs. In particular, edible coatings can serve as the barrier layer for the exchange of gas, water, and solutes to lower the respiration rate of fruits. These properties make food coatings highly promising in terms of reducing water loss and preventing spoilage [1]. A series of natural materials, including starch, gelatin, proteins, lipids, chitosan, and sodium alginate (SA), have been used to prepare food coatings for the postharvest preservation of fruits and vegetables [2]. Bananas are very popular worldwide due to their rich nutrients and wonderful taste. As a typical respiration climacteric fruit, a banana is prone to browning, softening, and rotting during a short period after picking. As a result, the shelf life of bananas is pretty short [3]. Therefore, it is of great necessity to develop efficient pretreatment strategies to prevent the browning of bananas to extend their shelf life. To solve this issue, several methods have been widely utilized, including storage at low temperatures and treatment with chemicals [4,5,6]. However, the quality of bananas could be affected and chemicals are harmful to human health [7]. Hence, it is highly desirable to develop safe and efficient methods to extend the shelf life of bananas. Attempts have been made to use chitosan, gum arabic, pectin, and maltodextrin edible coatings for the preservation of bananas. These strategies have been successful in extending the shelf life of bananas [8,9,10].

SA, a natural hydrophilic polysaccharide composed of β-D-mannuronic and α-L-guluronate, has attracted extensive attention due to its excellent physiochemical properties [11,12,13,14,15]. For instance, SA exhibits high compatibility, high transparency, excellent adhesion, and high flexibility and thus has been regarded as an ideal candidate for the preparation of films [16]. SA is designated as “generally recognized as safe” compared to other natural compounds. With a large number of reactive groups such as hydroxyl and carboxyl groups in its molecular chain, the structure and properties of SA are highly variable. This gives SA a unique advantage in the field of packaging development [17,18,19]. Incorporation of date palm extract into gelatin–sodium alginate membranes and successful extension of the shelf life of raw minced beef was reported by Khaoula Elhadef [20]. Milad Ranjbar incorporated red beet anthocyanin extract into SA film and successfully extended the shelf life of chicken tenders [21]. Zhuang Zhuang Qiu wrapped sausages with SA films and different lotus root powders and successfully extended the shelf life of sausages [22]. In addition, SA also displays excellent biodegradability, outstanding moisture resistance, and high capability to isolate oxygen. Owing to these features, SA shows great advantages in fruit preservation. To further enhance the performance of SA coatings, additives, especially antibacterial agents such as plant extracts [23], phenolics [24], essential oils [25], and nanoparticles [26], are often incorporated into the coatings to prolong the shelf life of food products.

Sodium nitroprusside (SNP) is a compound consisting of sodium ions (Na), ferrous iron (Fe^2+^), and nitric oxide (NO). SNP releases NO for medicinal effects such as vasodilatation and regulation of wound healing, reduced mortality in heart failure, and treatment of severe hypertension. In 1974, it received approval from the Food and Drug Administration (FDA) for the treatment of severe hypertension [27]. In recent years, silver nitroprusside (Ag_2_[Fe(CN)_5_NO]) nanoparticles, referred to as AgNNPs, have shown potential for medical applications due to their excellent physicochemical properties, antimicrobial properties, and biocompatibility, such as accelerated wound healing in mice and use as wound dressings in combination with hydrophobic cotton [28,29]. Yingshan Jin also used peony extract to assist in the synthesis of AgNNPs for biomedical applications [30]. AgNNPs have not yet been reported as antibacterials in packaging and coating applications. The potential of AgNNPs use in food coatings to extend the shelf life of fruits remains to be investigated. Therefore, this study aimed to synthesize and characterize SA-AgNNPs films and to investigate their antibacterial and antibiofilm activities. The biocompatibility of SA-AgNNPs films was also determined by toxicity assays (in vitro hemolysis and cytotoxicity). In addition, the coating and preservation properties of SA-AgNNPs films on bananas were evaluated by measuring weight loss, total sugar content, TBC, and anti-browning properties.

## 2. Materials and Methods

### 2.1. Materials

Nutrient agar, nutrient broth, Muller–Hinton agar, Dulbecco’s Modified Eagle Medium (DMEM), and Fetal Bovine Serum (FBS) were procured from Difco Laboratories Fisher Scientific, Republic of Korea. CellomaxTM kits (cell viability assay kits) were obtained from MediFab, Seoul, Korea. Other chemicals used, such as sodium alginate, silver nitrate, and sodium nitroprusside, were purchased from Sigma-Aldrich. The bacteria strains, including *Bacillus cereus* (ATCC 14579), *Staphylococcus aureus* (ATCC 19095), *Listeria monocytogenes* (ATCC 15313), *Salmonella enterica* (ATCC 14028), and *Escherichia coli* (ATCC 43888), were obtained from the Korea Culture Center for Microbiology, Korea. Mouse fibroblast cell lines (NIH3T3) were obtained from the Korea Cell Line Bank (KCLB) (Seoul National University College of Medicine, Seoul, Republic of Korea).

### 2.2. Synthesis of AgNNPs

The AgNNPs were synthesized using an approach according to previously published research that was slightly modified [30]. In detail, 25 mL of silver nitrate (AgNO_3_) solution (5 mM) was mixed with 50 mL of sodium nitroprusside dihydrate (SNP) (5 mM). Subsequently, the mixture was vigorously stirred for 18 h at room temperature in the dark. The resultant solution turns pink, indicating the formation of AgNNPs. These obtained nanoparticles were purified through centrifugation at 4000 rpm for 20 min. The formed pellet was washed three times with distilled water and dried in an oven at 50 °C for 12 h.

### 2.3. Preparation of SA-AgNNPs Films

A total of 90 mL of 1% (*w*/*v*) aqueous SA solution was measured and 25.48 mg of AgNO_3_ and 89.39 mg of SNP were added to the aqueous SA solution. It was stirred vigorously for 18 h at room temperature and in dark conditions. The final 1 mL of SA-AgNNPs film-forming solution contained about 1.28 mg of AgNNPs. A total of 15 mL of the as-obtained film solution was added to a glass dish with a diameter of 90 mm. The dishes were dried in the oven at 37 °C for 18 h to obtain the SA-AgNNPs films. Pure SA film was used as a control.

### 2.4. Characterization of AgNNPs, SA Films, and SA-AgNNPs Films

A Field Emission Transmission Electron Microscope (FE-TEM) (JEOL-JSM 1200EX, Japan) and energy-dispersive X-ray spectroscopy (EDX) (JEM-2100F, JEOL, Tokyo, Japan) were utilized to analyze the shape, size, and elemental composition of AgNNPs. The cross-sectional morphology and elemental composition of SA films and SA-AgNNPs films were observed under a Field Emission Scanning Electron Microscope (FE-SEM) (Hitachi S-4800, Tokyo, Japan) and EDX (JEM-2100F, JEOL, Tokyo, Japan) [31]. Photos of the films on white paper with a logo were taken with a digital camera (KDL-32 W600D, Sony, Tokyo, Japan). A double-beam, UV–visible spectrophotometer (Specord 210, Analytik Jena, Jena, Germany) was employed for light transmittance estimation with a scanning range from 200 to 700 nm [32]. X-ray diffraction (XRD) (X’pert-pro MPD-PANalytical, Worcestershire, UK) analysis was performed to obtain the crystal structures of AgNNPs, SA films, and SA-AgNNPs films. The functional properties of AgNNPs, SA films, and SA-AgNNPs films were characterized through attenuated total reflectance–Fourier transform infrared spectroscopy (ATR-FTIR) (FTIR, PerkinElmer Paragon 500, Waltham, MA, USA). To determine the particle size, dispersion, and ζ-potential of AgNNPs and SA-AgNNPs films, 200 µg of AgNNPs was dispersed in 1 mL of phosphate-buffered saline (PBS) and 1 mL of SA-AgNNPs pre-film-forming solution in PBS (5 mL) and sonicated for 30 s. This mixture was then analyzed by dynamic light scattering (DLS) and electrolytic light scattering (ELS) analysis (Malvern PANalytical, Worcestershire, UK).

### 2.5. Well Diffusion Assay of SA-AgNNPs Films

The inhibitory activity of SA-AgNNPs films against *B. cereus*, *S. aureus*, *L. monocytogenes*, *S. enterica*, and *E. coli*. was determined by the agar well diffusion method [33]. For bacterial inoculation, a sterile cotton swab was dipped in bacterial solution (10^7^ CFU/mL) and evenly cultured on the Mueller–Hinton agar (MHA) plates. Prepared SA-AgNNPs films were cut into small pieces with a diameter of 6 mm, which were further placed onto the agar plate containing each bacteria mentioned above. Subsequently, the plates were incubated at 37 °C for 8 h. After that, the sizes of the inhibition zones were measured and calculated to evaluate the antibacterial activity of the SA-AgNNPs films. Tetracycline hydrochloride (TCH) was used as the positive control.

### 2.6. MIC and MBC Assays of SA-AgNNPs Films

The minimum inhibitory concentration (MIC) and minimum bactericidal concentration (MBC) of SA-AgNNPs films were evaluated using the micro-broth dilution method based on CLSI standards [34]. Briefly, the pathogens were initially cultured in Mueller–Hinton broth (MHB) for 24 h. Then, *B. cereus*, *S. aureus*, *L. monocytogenes*, *S. enterica*, and *E. coli* were inoculated in 96 wells containing MHB. To obtain different concentrations, SA-AgNNPs films (*w*/*v*) were dissolved in distilled water and serially diluted. A volume of 100 μL of SA-AgNNPs films of different concentrations was added to 100 μL of pathogens, which was subsequently incubated at 37 °C for 24 h. The absorbance at 600 nm after incubation for different time periods (0 h, 2 h, 4 h, 6 h, 12 h, 24 h) was determined using a UV spectrophotometer (SpectraMax^®^ABS Plus, Molecular Devices, San Jose, CA, USA). Growth kinetic curves were prepared and used to calculate the MIC and MBC.

### 2.7. Anti-Biofilm Assay of SA-AgNNPs Films

The anti-biofilm properties of SA-AgNNPs films were evaluated in *B. cereus*, *S. aureus*, *L. monocytogenes*, *S. enterica*, and *E. coli* according to the previously reported literature [35]. For the biofilm assays, bacterial pathogens inoculated with MHB were poured into 96-well plates and incubated with different concentrations of SA-AgNNPs films for 24 h at 37 °C. The medium containing planktonic bacterial cells was gently removed from the plates and the adherent bacterial cells were rinsed three times with PBS. Subsequently, the adherent cells were fixed on the wells for 30 min using 0.1% crystal violet solution (200 μL). In addition, the plates were washed thoroughly with PBS to remove the unbounded crystal violet. The plates were then inverted for 60 min at room temperature. To quantify the biofilm, the bound crystalline violet was dissolved in ethanol (200 μL) with the affixed bacterial cells and evaluated at 590 nm.

### 2.8. Cytotoxicity Assays

The cytotoxicity assays of AgNNPs, SA films, and SA-AgNNPs films were evaluated using WST assay kits (CelloMax, Seoul, Korea). Briefly, 100 μL of NIH3T3 cells (1 × 10^4^) was added into a 96-well plate which was then incubated at 37 °C at a CO_2_ concentration of 5% for 24 h. After incubation, 10 μL of AgNNPs, SA films, and SA-AgNNPs films with different concentrations were added to the plates. AgNNPs and SA films were diluted in the same way as SA-AgNNPs films in Section 2.6 and incubated for 12 h. Subsequently, 10 μL of WST reagent was added for each well in the plate. The absorbance at 450 nm was measured using a UV spectrophotometer after a reaction time of 30 min, with the cytotoxicity then calculated [36,37].

### 2.9. Hemolysis Assay

The hemolysis assay was carried out according to a previously reported method [38]. Specifically, 1 mL of solution of sheep blood erythrocytes was added into 9 mL of PBS. The mixture was centrifuged at 2000 rpm for 10 min, followed by washing with PBS three times and dispersion into PBS. Subsequently, different concentrations (1.9–500 μg/mL) of AgNNPs, SA films, and SA-AgNNPs films were mixed with the sheep blood erythrocytes PBS solution at a volume ratio of 1:1. After incubation at 37 °C for 1 h, the reacted solutions were centrifuged at 2000 rpm for 10 min. A UV spectrophotometer was utilized to measure the absorbance of the supernatant at 545 nm. It is worth noting that Triton X-100 (1% *v*/*v*) was used as a positive control, whereas PBS was used as a negative control. The hemolysis ratio was calculated according to the following equation:$$\mathrm{Hemolysis}\;(\%)=\frac{A_{0}-A_{2}}{A_{1}-A_{2}}\;\times \;100\%$$
where A_0_, A_1_, and A_2_ represent the absorbance of the experimental group, the positive group, and the negative group, respectively.

### 2.10. Banana Preservation

To investigate the effect of SA-AgNNPs films on the freshness of bananas, SA films and SA-AgNNPs films were coated onto bananas as a comparison. Fresh bananas with the same size, ripeness, and weight were bought from a local market in Chuncheon City, Korea. Bananas were washed with water and then dried at room temperature before then being divided into three groups: an uncoated group, SA-treated bananas, and SA-AgNNPs-treated bananas. For the preparation of SA-treated bananas and SA-AgNNPs-treated bananas, bananas were respectively immersed in SA and SA-AgNNPs pre-film-forming solutions for 30 s. The treated bananas were dried at room temperature to form uniform surface films. The treated bananas were then stored at room temperature to record the changes in appearance and physicochemical properties [10].

### 2.11. Weight Loss of Bananas during Preservation

The weight loss of bananas during preservation was measured and calculated using the following equation:$$\mathrm{Weight}\;\mathrm{loss}=\frac{B_{0}-B_{1}}{B_{0}}\times 100\%$$
where B_0_ refers to the initial mass of the banana samples and B_1_ refers to the weight of the stored bananas at a certain time. Weight loss of the bananas was calculated on days 1, 3, 5, 7, and 9 [39].

### 2.12. Measurements of Total Sugar Content and Total Bacterial Colonies in Bananas

About 25 g of banana containing pulp and peel was ground with a small amount of water into a slurry. The slurry was then poured into a volumetric flask and water was added to a constant volume of 250 mL. After ultrasonication, 1 mL of the supernatant was taken and diluted into 100 mL of the mixture. A volume of 50 μL of the diluted sample was mixed with 30 μL of 5% phenol (*w*/*v*) and 150 μL of concentrated sulfuric acid in turn. The mixture was heated in a water bath at 90 °C for 5 min, followed by determining the absorbance at 590 nm. A series of glucose solutions with different concentrations were utilized to set a standard curve to obtain the sugar content of banana samples [40]. Also, the TBC values of the stored bananas were measured on days 1, 3, 5, 7, and 9 in accordance with previously reported works [41,42].

### 2.13. Statistical Analyses

GraphPad Prism 9 was used to plot the data, and the collection of experimental data was repeated three times. All data are expressed as means ± standard deviations. The one-way analysis of variance was performed using IBM SPSS statistical software to determine the significance between the two groups (*p* < 0.05).

## 3. Results and Discussion

### 3.1. Characterizations of AgNNPs, SA Films, and SA-AgNNPs Films

#### 3.1.1. TEM and SEM Analysis of AgNNPs, SA Films, and SA-AgNNPs Films

TEM images of AgNNPs with elemental mappings and elemental spectra are shown in Figure 1a,b. It can be observed that the prepared nanoparticles exhibit a cubic morphology with high dispersion and an average diameter of about 400 nm. EDX spectrum of the nanoparticles indicated that the elements Ag, Fe, C, N, and O are present in the prepared nanoparticles. Among the five elements, Ag content is the highest compared to the other elements, followed by C, Fe, N, and O. Moreover, the elemental mapping analysis showed that all five elements, namely Ag, Fe, C, N, and O, are uniformly distributed among the nanoparticles, with Ag and Fe also found to be mainly concentrated on AgNNPs, which is consistent with previous research [30]. These results demonstrated that AgNNPs were successfully prepared. Furthermore, an FE-SEM was used to characterize the morphology of SA films and SA-AgNNPs films. As shown in Figure 2a–f, SA films display a uniform and smooth surface structure. After incorporation with AgNNPs, it can be seen that plenty of small particles are well distributed on the surface of SA films. The results indicate that AgNNPs possess a high affinity to SA films through intermolecular interactions [43]. The EDX spectrum of the SA-AgNNPs films in Figure 2g shows that the films contain the C, O, Fe, Na, and Ag elements, demonstrating that the SA-AgNNPs films were successfully prepared.

#### 3.1.2. Physical Appearance and Optical Properties of SA Films and SA-AgNNPs Films

The appearance of food packaging has a great impact on consumer interest in food products. Photographs of the SA film and SA-AgNNPs films are presented in Figure 3a. Both films were transparent. The SA films were colorless, while SA-AgNNPs films exhibited a brownish-yellow color due to the presence of AgNNPs. To further demonstrate this, the SA film and SA-AgNNPs films were placed on the top of the logo of Kangwon National University (KNU), the results indicate that both films exhibit high transparency. Additionally, the concentration of SA was varied in fabricated SA-AgNNPs films. It can be observed that the concentration of SA has negligible effects on the transparency of films (Appendix A). In transparent food packaging, light reduces food quality and accelerates spoilage. The UV-vis spectrophotometric light-blocking ability of the films is shown in Figure 3b. Therefore, food packaging with light-blocking properties protects food [10]. The UV–vis spectrophotometric light-blocking ability of the films is shown in Figure 3b. SA films showed high transmittance in the 200–280 nm (UVC) light region, whereas no light passed through the SA-AgNNPs films at a UV wavelength below 280 nm, indicating complete prevention. In addition, SA-AgNNPs films partially blocked UVB (280–315 nm) and UVA (315–400 nm) light, and the transmittance in these regions was much lower than in SA films. In particular, at 280 nm, the transmittance of the SA-AgNNPs films and SA films were 0.53% and 62.41%, respectively. The maximum transmittance of the SA films at 700 nm was 77.75%, while the transmittance of SA-AgNNPs films at the same wavelength was reduced to 54.43%. This barrier property of SA-AgNNPs films to UV and visible light may be due to the scattering of light by obstacles in the passage of light or by nanoparticles dispersed in the film matrix [44]. According to our results, SA-AgNNPs films can be used for UV protection packaging.

#### 3.1.3. XRD and FTIR Analysis of AgNNPs, SA Films, and SA-AgNNPs Films

XRD and FTIR spectra were analyzed to verify the synthesis of SA-AgNNPs films. As shown in Figure 4a, AgNNPs exhibited a series of characteristic peaks at 13.88° and 19.18°. Such peaks can be referred to as the diffraction peaks of the (010) and (100) planes in the face-centered cubic structure, consistent with the displayed peak positions of XRD spectra of simulated AgNNPs (JCPDS No.: 052-0366), thus showing the successful synthesis of AgNNPs [45]. In addition, SA films exhibited two typical broad peaks at 13.46° and 21.59°, with the peak at 21.59° originating due to the amorphous nature of SA [46]. SA-AgNNPs films displayed peaks at 13.23° and 19.18°, which are consistent with those of AgNNPs. Alongside the XRD characterization, FTIR spectra of AgNNPs, SA films, and SA-AgNNPs films are illustrated in Figure 4b. AgNNPs displayed two intense bands at about 2175 cm^−1^ and 1931 cm^−1^, corresponding to the stretching vibrations of the C≡N and N=O groups, respectively [47,48,49]. SA films exhibited a series of peaks at 3257 cm^−1^, 1593 cm^−1^, and 1407 cm^−1^, which can be ascribed to the vibrations of O-H and the asymmetric and symmetric vibrations of -COO-, respectively [50]. SA-AgNNPs films exhibited an absorption peak at 1937 cm^−1^ near the characteristic peaks of AgNNPs. However, compared with pure AgNNPs, the intensity of the peak at 1937 cm^−1^ was much weaker in SA-AgNNPs films. This is mainly due to the fact that AgNNPs are encapsulated by SA [51]. Moreover, compared to pure SA powders, there were obvious decreases in the intensities of peaks referring to the vibrations of O-H and the asymmetrical and symmetric vibrations of -COO-. This is because the strong interaction between AgNNPs and SA weakens the intermolecular hydrogen bonding between SA molecules. These results indicate that AgNNPs were successfully incorporated into the films.

#### 3.1.4. Zeta Size and Zeta Potential Analysis of AgNNPs and SA-AgNNPs

Size and surface potentials of nanomaterials are two important parameters to assess the efficiency of nanoparticle cellular uptake and cellular permeability. Therefore, DLS analysis was performed to characterize the size distribution and potentials of AgNNPs and SA-AgNNPs. As shown in Table 1, as well as Appendix A, AgNNPs exhibited an average hydrodynamic size of 541.43 ± 15.19 nm, while SA-AgNNPs exhibited an average size of 365.60 ± 30.75 nm. The smaller average size of SA-AgNNPs is because the coating of sodium alginate on the surface of the nanoparticles limits the growth of the nanoparticles. The polydispersity index (PDI) values of AgNNPs and SA-AgNNPs were 0.368 ± 0.03 and 0.471 ± 0.16, respectively. A PDI value < 0.5 has been considered to represent that the material was homogenous and stable [52]. A lower PDI value represents higher size homogeneity. Therefore, it can be inferred from the PDI values that the size distribution of AgNNPs is more uniform than that of SA-AgNNPs. This can be explained by the fact that nanoparticles with a large size are prone to showing a narrow size distribution. Moreover, it can be observed from Table 1 and Appendix A that the surface potentials of AgNNPs and SA-AgNNPs are −26.97 ± 0.21 mV and −58.40 ± 2.36 mV, respectively. The potential of SA-AgNNPs was more negative than that of AgNNPs due to the capping of SA on the surface of AgNNPs considering the carboxyl group in SA molecules [53]. Generally, the results indicate that both AgNNPs and SA-AgNNPs exhibit high dispersion.

### 3.2. Antibacterial Activity of SA-AgNNPs Films

#### 3.2.1. Well Diffusion Assay of SA-AgNNPs Films

Bacterial foodborne pathogens, including *B. cereus*, *S. aureus*, *L. monocytogenes*, *S. enterica*, and *E. coli*, were used as the microorganism models to investigate the antibacterial activity of SA-AgNNPs films. First, the SA concentration and the concentration ratio of AgNO_3_ to SNP used to synthesize the AgNNPs in the films were optimized to obtain the best antibacterial properties for further experiments, as shown in Appendix A and Appendix A. Specifically, AgNO_3_ and SNP were added to 0.25%, 0.5%, 1%, and 2% (*w*/*v*) SA aqueous solutions using concentration ratios of 1:1, 1:2, and 2:1, respectively. The antibacterial properties of the resulting samples were examined. According to Appendix A, the 1% (*w*/*v*) aqueous solution of SA added at a concentration ratio of 1:2 between AgNO_3_ and SNP was the optimal synthesis ratio for the antibacterial properties. Therefore, this ratio was chosen for subsequent experiments. The optimized SA-AgNNPs films were then used for the subsequent antibacterial experiments, and the results are shown in Figure 5. Unsurprisingly, the SA-AgNNPs films showed significant antibacterial activity against all tested bacterial pathogens compared to the pure SA films (*p* < 0.05). Also, AgNNPs displayed strong activities that inhibit bacteria. Furthermore, the diameters of the inhibition zones of different bacteria were measured and are shown in Table 2. Pure SA films exhibited no inhibition effects on the five bacteria strains, while the diameters of the inhibition zones caused by SA-AgNNPs films were 14.50 ± 0.5 mm (*B. cereus*), 14.67 ± 0.58 mm (*S. aureus*), 16.83 ± 0.29 mm (*L. monocytogenes*), 19.67 ± 0.58 mm (*S. enterica*), and 15.67 ± 0.58 mm (*E. coli*), respectively. These results suggest that AgNNPs have a significant effect on the antibacterial activity of the films (*p* < 0.05). Several works have reported that AgNNPs themselves possess strong antibacterial properties [28]. The possible mechanism of such a property involves the inducement of oxidative stress responses to cause lipid peroxidation, which can further lead to membrane damage to kill bacteria. In addition, the inhibition zone of *S. enterica* was larger than other pathogens, which is mainly because Gram-positive and Gram-negative bacteria have different cell structures [54].

#### 3.2.2. MIC and MBC of SA-AgNNPs Films

Bacterial growth experiments and determination of MIC and MBC values were carried out using well diffusion and 96-well plate turbidity tests, and the results are shown in Figure 6a and Table 3. Different concentrations of SA-AgNNPs films were used to determine bacterial growth at different time intervals up to 24 h. The results showed that for *E. coli* and *L. monocytogenes*, low concentrations of SA-AgNNPs films did not inhibit their growth, while high concentrations completely limited growth. A similar pattern was seen in *S. enterica*, where 250 μg/mL of SA-AgNNPs films was sufficient to restrict complete growth. Thus, MICs and MBCs of >500, 250, 500, 500, and >500 μg/mL were observed for SA-AgNNPs films against all of these five bacteria. These results support the significant broad-spectrum antibacterial activity of SA-AgNNPs films. The difference in antibacterial activity against different bacteria may be attributed to their different surface structures [55].

#### 3.2.3. Anti-Biofilm Assay of SA-AgNNPs Films

The ability of SA-AgNNPs films to inhibit biofilms was further investigated. Bacterial biofilms are three-dimensional structures of bacterial populations formed by the adhesion of bacterially secreted extracellular polymers to object contact surfaces [56]. Here, the anti-biofilm-forming effect of SA-AgNNPs films was evaluated through the use of crystal violet staining assays. As shown, SA-AgNNPs films exhibited biofilm inhibitory effects on all of these pathogens compared to the control group. Additionally, there was a dose-dependent relationship between the biofilm inhibition rate of the five bacteria and the concentration of SA-AgNNPs films (Figure 6b). For example, as the concentration of SA-AgNNPs films increased, the biofilm inhibition rate on the bacteria gradually increased. With a concentration of 500 μg/mL, the inhibition ratios of *B. cereus*, *S. aureus*, *L. monocytogenes*, *S. enterica*, and *E. coli*. were more than 60%, which is consistent with previous research [30]. Overall, these biofilm inhibition results indicate that SA-AgNNPs films had a better antibacterial effect.

### 3.3. Safety Assessment of AgNNPs, SA Films, and SA-AgNNPs Films

#### 3.3.1. Cytotoxicity Assays of SA-AgNNPs Films

Cytotoxicity assays are usually conducted to assess the cellular response to potentially toxic substances [57]. Therefore, to evaluate the biocompatibility of SA-AgNNPs films in practical applications, the NIH3T3 cell was used as the model cell and cell viability was tested using MTT to perform cytotoxicity assays. The results are displayed in Figure 7a. Cell viability was 77.66% at the AgNNPs concentration of 7.8 µg/mL. As the concentration of AgNNPs increased, NIH3T3 cell viability decreased, indicating that a high concentration of AgNNPs is toxic to NIH3T3 cells. However, SA-AgNNPs films exhibited much lower toxicity toward NIH3T3 cells. No obvious toxicity was observed at SA-AgNNPs concentrations ranging from 1.9 μg/mL to 62.5 μg/mL, where cell viability was maintained above 90%. A possible explanation is that because the AgNNPs are coated by SA molecules, fewer AgNNPs are exposed to interact with cells [58]. These results indicate that SA-AgNNPs films exhibit high biocompatibility [59].

#### 3.3.2. Hemolysis Assays of SA-AgNNPs Films

Hemolysis assays were conducted to further evaluate the biocompatibility and safety of SA-AgNNPs films. As shown in Figure 7b,c, supernatants of both the SA film- and SA-AgNNPs film-treated groups were transparent with a slightly yellow color. Even though the concentration of SA films and SA-AgNNPs films was as high as 500 μg/mL, the hemolysis ratios of the SA film- and SA-AgNNPs film-treated groups were only 0.33% and 0.46%, respectively. Such hemolysis ratios were much lower than the permissible limit (5%) [60]. However, for AgNNPs, the hemolysis ratio increased with an increasing concentration of nanoparticles. Obvious hemolysis was observed when the AgNNPs concentration was relatively high. As a control group, Triton-X was also used to induce hemolysis, and a significantly red color was observed. This suggests that the red blood cells were severely damaged. All these results suggest that SA-AgNNPs films show negligible effects on red blood cells and exhibit excellent safety. The low hemolysis and high biosafety guarantee that SA-AgNNPs films can be utilized as a harmless material for food packaging.

### 3.4. Application for Banana Preservation

#### 3.4.1. Changes in Visual Appearance

Effective storage of bananas during their production, transportation, and distribution remains a challenging task. Herein, SA-AgNNPs films were coated onto the surface of bananas to investigate their performance in terms of banana preservation. All the bananas with and without a coating were placed at 25 °C, and their appearance at different time points was recorded and is shown in Figure 8a. In the beginning, all bananas possessed intact peels and a uniform color of yellowish green. After storage for 5 days, large areas of brown color can be observed on the surface of bananas without coatings. However, only small amounts of black spots appeared on the surface of both SA-coated bananas and SA-AgNNPs-coated bananas. Furthermore, after 9 days, more black senescent spots were observed on uncoated bananas, SA-coated bananas, and SA-AgNNPs-coated bananas. Compared to uncoated bananas, SA-coated bananas and SA-AgNNPs-coated bananas displayed a much better appearance. This is due to the SA-based coatings providing efficient barriers against moisture and oxygen, which weakens the respiration of bananas to slow down ripening [61].

#### 3.4.2. Changes in Weight Loss

During storage, evaporation of water and conversion of glucose to CO_2_ in respiratory cycles leads to significant weight loss. Therefore, weight loss is recognized as one of the most critical parameters for predicting postharvest fruit deterioration [62]. Metabolic reactions are responsible for the fruit ripening process, which is expressed through the weight loss of banana fruits [63]. A suitable coating can act as an effective barrier against water escape and the penetration of oxygen and carbon dioxide into the banana, thus reducing water loss and slowing down metabolic reactions and respiratory processes [64]. The weight loss of SA-AgNNPs-coated, SA-coated, and uncoated bananas is shown in Figure 8d, weight loss for all treatments increased gradually with storage time, and it can be seen that uncoated bananas exhibited rapid weight loss and reached 27.19% at the end of storage. Meanwhile, the weight loss of coated bananas during storage was slowed down. As observed by Pinzon, chemical interactions between compounds in the coated films reduce water vapor permeability and water loss [65]. At day 9, weight loss was 27.19%, 11.98%, and 11.22% for the uncoated, SA-coated, and SA-AgNNPs-coated bananas, respectively. It can be noticed that the weight loss further decreased when AgNNPs were added to the SA edible coating. The decrease in weight loss upon incorporation of AgNNPs in the SA coating may be attributed to the antibacterial properties of AgNNPs [66]. The results showed that the SA-AgNNPs coating provided the most effective freshness preservation performance for bananas during storage compared to uncoated and SA-coated fruits.

#### 3.4.3. Changes in Total Soluble Sugars

G. Veerapandi stated that total soluble sugar is one of the most important criteria for determining the postharvest quality of bananas [67]. Total soluble sugars in treated and untreated bananas were assessed during storage, as shown in Figure 8b,c. In general, the total sugar content of all bananas increased gradually as the storage period increased. The total sugar content of postharvest bananas tended to increase due to the hydrolysis of starch to sugars such as glucose, sucrose, and fructose [62]. For banana skin, at day 9, the total sugar content was lower in the SA-coated and SA-AgNNPs-coated groups compared to the uncoated group. This was consistent with the pictures of bananas during the storage period, where the banana peels of the uncoated bananas were all blackened and wrinkled with water loss whereas the bananas of the SA-coated and SA-AgNNPs-coated groups were still glossy (Figure 8a). For banana pulp, the total sugar content of the three samples was similar in the first three days, while on the fifth day of storage, the total sugar content of the SA-coated and SA-AgNNPs-coated groups was lower compared with that of the uncoated group (Figure 8c). From day 5 to day 9, the total sugar content of the uncoated group and SA film group showed a similar trend of increase, while the total sugar content of bananas in the SA-AgNNPs-coated group increased at a lower rate. The increase in total sugar content was probably due to the conversion of starch to sugar, and the decline in starch content accompanied by an increase in total sugar is a typical feature of postharvest changes in fruits at the leaping stage [9]. Starch hydrolysis and sugar synthesis in bananas are usually completed at full ripening. The rapid increase and subsequent rapid decrease in total sugar content in the uncoated group and SA-coated group on days 5–9 could be attributed to the faster ripening process of converting starch to sugar. In contrast, the increase in total sugar content at a slower rate in the SA-AgNNPs-coated group could be attributed to the effect of delaying the ripening process in bananas [68]. It is worth mentioning that surface coating can reduce respiration frequency and thus reduce the hydrolysis of starch [9].

#### 3.4.4. Changes in TBC

Microbiological analysis is an important factor in evaluating the effectiveness of banana preservation. The effect of different packaging materials on the bacterial colony counts of banana samples is shown in Table 4. The colony counts in the peel and pulp of all banana samples showed an increasing trend with increased storage time. The total number of colonies of bananas increased very little at the beginning of the storage period. After 9 days of storage, the total number of bacterial colonies in banana skin increased to 8.96 × 10^5^ CFU/g in the uncoated group. TBC values in the banana skin of the SA-treated group and SA-AgNNPs-treated groups were 1.09 × 10^5^ CFU/g and 1.13 × 10^3^ CFU/g, respectively. Additionally, TBC values in the banana pulp of the uncoated group and SA-treated group were 3 × 10^3^ CFU/g and 1.64 × 10^2^, respectively. However, the TBC value in the banana pulp of SA-AgNNPs-treated groups was only 51 CFU/g. Compared with the other two groups, the total number of bacterial colonies in the SA-AgNNPs-coated groups was much lower. Such results demonstrate that SA-AgNNPs film showed good performance in inhibiting bacterial growth. This effect is consistent with the results of the film’s antibacterial capacity, indicating that SA-AgNNPs film with high antibacterial activity can effectively inhibit the growth of banana microorganisms. This favorable inhibition can be attributed to the robust antibacterial effect of AgNNPs in SA films.

## 4. Conclusions

In summary, AgNNPs were incorporated into SA coatings to prevent bananas from spoilage. The successful preparation of SA-AgNNPs films was verified by SEM, XRD, FTIR, and zeta analysis. The synthesized SA-AgNNPs showed an average size of 365.60 ± 30.75 nm and surface potentials of −58.40 ± 2.36 mV. SA-AgNNPs films also have good UV blocking properties. Antibacterial assays indicated that SA-AgNNPs films exhibited high antibacterial activity against five pathogens. Cytotoxicity assays and hemolysis assays suggested that SA-AgNNPs films displayed high biosafety at concentrations ranging from 1.9 μg/mL to 62.5 μg/mL. Further, the SA-AgNNPs films were utilized for banana preservation. SA-AgNNPs films can serve as a physical barrier to slow down the ripening of bananas and reduce the weight loss of bananas. Additionally, SA-AgNNPs films can also act as an antibacterial material to inhibit the growth of microorganisms, thus reducing the total bacterial colonies of banana samples. These two advantages synergistically help to extend the shelf life of bananas. Compared with SA-coated and uncoated bananas, SA-AgNNPs-coated bananas exhibited the lowest weight loss and the lowest total bacterial colonies values. These results demonstrate that SA-AgNNPs films hold great promise in the postharvest preservation of fruits and vegetables and may provide an efficient strategy to prevent the spoilage of tropical agricultural products during the circulation and storage process. In subsequent works, deep mechanisms related to the effects of SA-AgNNPs films on banana preservation should be investigated. Moreover, applications of SA-AgNNPs films in the preservation of other fruits and vegetables should be explored to investigate the generalizability of SA-AgNNPs films.

## Figures and Tables

**Figure 1 nanomaterials-14-00292-f001:**
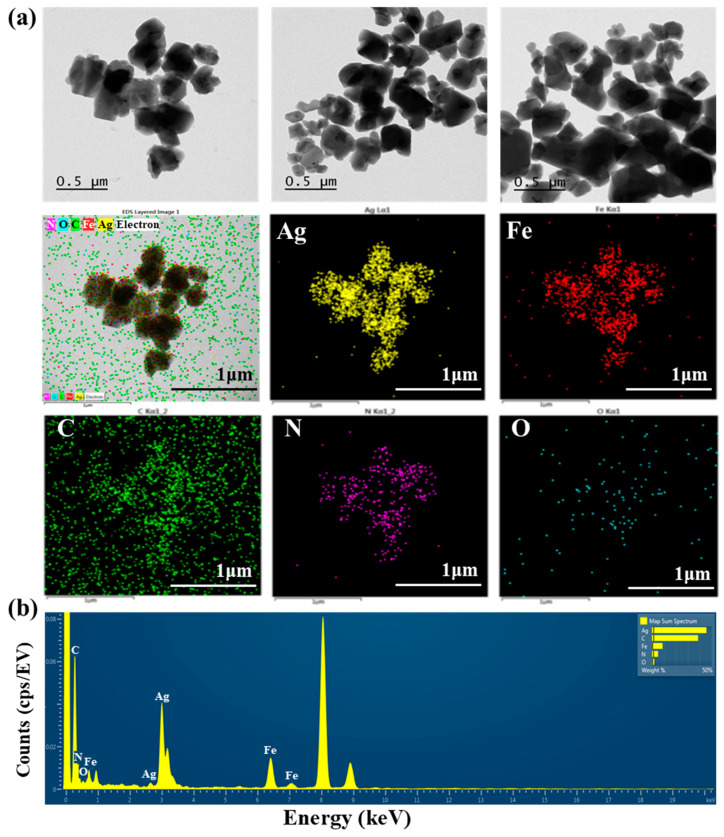
Transmission electron microscopy analysis of AgNNPs with elemental mappings: images of silver (yellow), iron (red), carbon (green), nitrogen (purple), and oxygen (blue) (**a**) and the EDX spectrum (**b**).

**Figure 2 nanomaterials-14-00292-f002:**
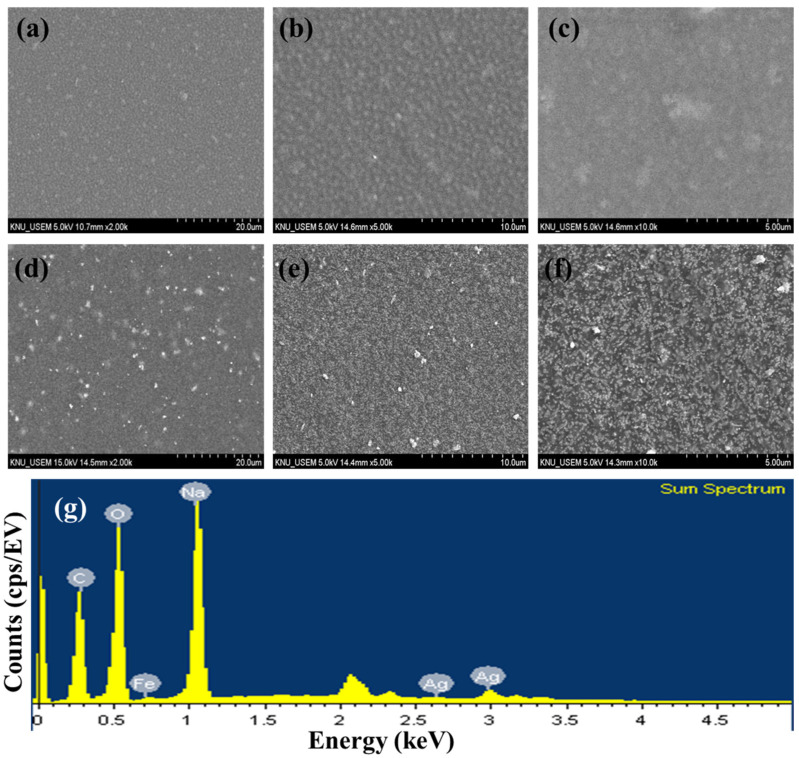
Scanning electron microscopy images of SA films (**a**–**c**) and SA-AgNNPs films (**d**–**f**) and the EDX spectrum (**g**).

**Figure 3 nanomaterials-14-00292-f003:**
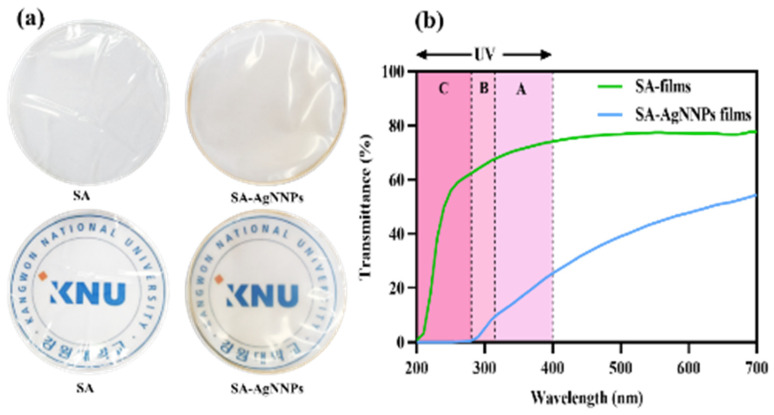
Digital photos and photos with a background of the school logo (**a**) and the UV–visible light transmission profile (%) (**b**) of SA films and SA-AgNNPs films.

**Figure 4 nanomaterials-14-00292-f004:**
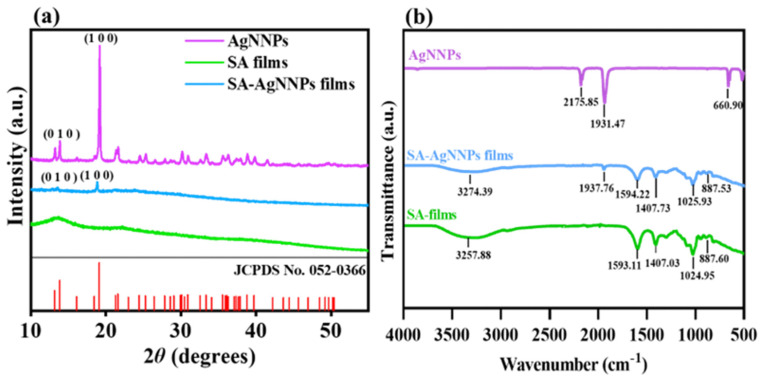
X−ray diffraction analysis (XRD) (**a**) and Fourier transform infrared spectroscopy (FTIR) spectra (**b**) of AgNNPs, SA films, and SA-AgNNPs films.

**Figure 5 nanomaterials-14-00292-f005:**
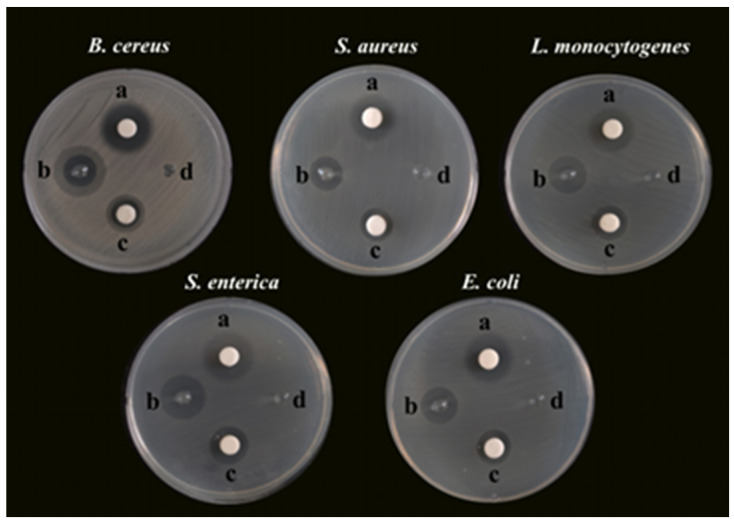
Photographs of the glass plates for bacteria incubation: a, TCH, 50 μg; b, SA-AgNNPs films, 6 mm; c, AgNNPs, 50 μg; d, SA films, 6 mm.

**Figure 6 nanomaterials-14-00292-f006:**
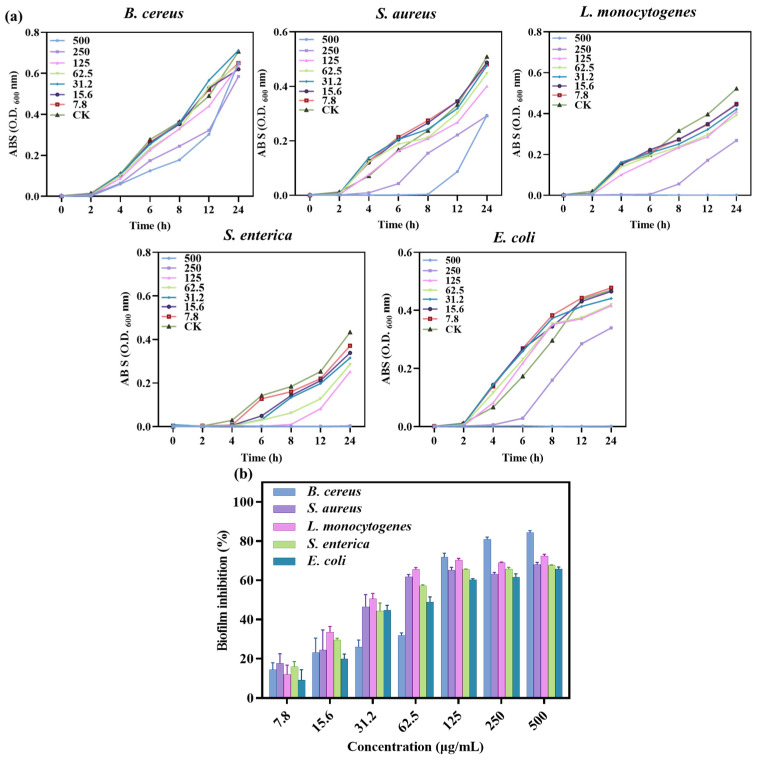
Growth curves of different bacterial pathogens with different concentrations of SA-AgNNPs films (**a**) and the biofilm inhibition ratio of SA-AgNNPs films against different pathogens (**b**).

**Figure 7 nanomaterials-14-00292-f007:**
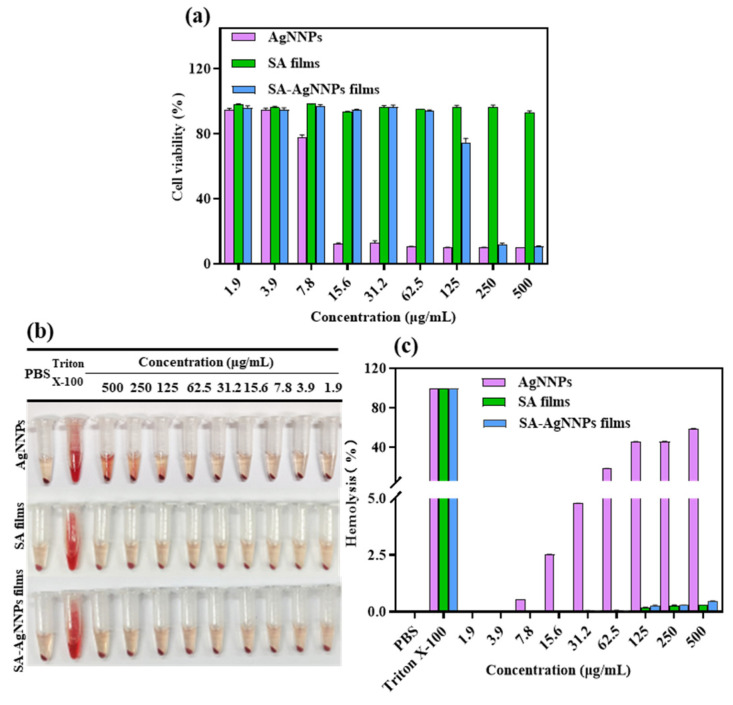
Cytotoxic effect of AgNNPs, SA films, and SA-AgNNPs films against the NIH3T3 cell line (**a**) and the hemolytic properties of AgNNPs, SA films, and SA-AgNNPs films (**b**,**c**).

**Figure 8 nanomaterials-14-00292-f008:**
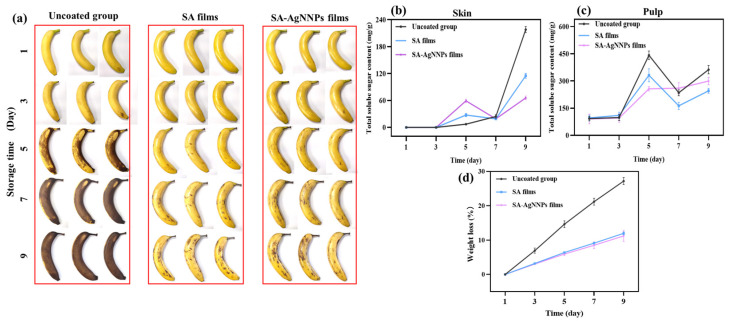
Appearance changes (**a**), total soluble sugar content in the skin (**b**), total soluble sugar content in pulp (**c**), and weight loss (**d**) of uncoated bananas, SA-coated bananas, and SA-AgNNPs-coated bananas when stored at 25 °C.

**Table 1 nanomaterials-14-00292-t001:** Zeta potentials, size, and particle size distribution (PDI) of AgNNPs and SA-AgNNPs.

Sample	Size (d. nm)	Zeta Potential (mV)	PDI
AgNNPs	541.43 ± 15.19	−26.97 ± 0.21	0.368 ± 0.03
SA-AgNNPs	365.60 ± 30.75	−58.40 ± 2.36	0.471 ± 0.16

**Table 2 nanomaterials-14-00292-t002:** The zones of inhibition against different bacterial pathogens for AgNNPs, SA films, and SA-AgNNPs films compared to tetracycline-hydrochloride (TCH).

Bacterial Strain	Zone of Inhibition (mm)			
	AgNNPs (50 μg)	SA Films (6 mm)	SA-AgNNPs Films (6 mm)	TCH (50 μg)
*B. cereus*	12.67 ± 0.29 ^c^	0 ^d^	14.50 ± 0.5 ^b^	20.67 ± 0.58 ^a^
*S. aureus*	13.33 ± 0.58 ^c^	0 ^d^	14.67 ± 0.58 ^b^	17.67 ± 0.58 ^a^
*L. monocytogenes*	16.33 ± 0.29 ^b^	0 ^c^	16.83 ± 0.29 ^b^	17.67 ± 0.58 ^a^
*S. enterica*	16.33 ± 0.58 ^c^	0 ^d^	19.76 ± 0.58 ^b^	21 ± 0 ^a^
*E. coli*	14.83 ± 0.29 ^b^	0 ^c^	15.67 ± 0.58 ^b^	19 ± 1 ^a^

Different letters indicate significant differences between the cross-sectional samples (*p* < 0.05).

**Table 3 nanomaterials-14-00292-t003:** MIC/MBC of SA-AgNNPs films against *B. cereus*, *S. aureus*, *L. monocytogenes*, *S. enterica*, and *E. coli*.

	MIC/MBC (µg/mL)				
SA-AgNNPs films	*B. cereus*	*S. aureus*	*L. monocytogenes*	*S. enterica*	*E. coli*
	>500	>500	500	250	500

**Table 4 nanomaterials-14-00292-t004:** The total bacterial colonies (TBC) of bananas when stored at 25 °C.

Factors	Treatments		Storage Intervals (Days)/ Storage Intervals (Days)/TBC (CFU/g)				
			1	3	5	7	9
TBC	Skin	Uncoated group	2 × 10^2^	2.3 × 10^3^	2 × 10^3^	1.5 × 10^5^	8.96 × 10^5^
		SA films	2.5 × 10^2^	1.7 × 10^3^	1.8 × 10^3^	3.6 × 10^4^	1.09 × 10^5^
		SA-AgNNPs films	1.3 × 10^2^	9.2 × 10^2^	8.9 × 10^2^	5.4 × 10^2^	1.13 × 10^3^
	Pulp	Uncoated group	0	0	3.3 × 10^2^	5 × 10^2^	3 × 10^3^
		SA films	0	0	16	100	1.64 × 10^2^
		SA-AgNNPs films	0	0	10	50	51

## Data Availability

Data are contained within the article or the Appendix A.

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
