# Peer review of "Development and Analysis of Silver Nitroprusside Nanoparticle-Incorporated Sodium Alginate Films for Banana Browning Prevention"

_nanomaterials, 2024, doi:10.3390/nano14030292_

Round 1
Reviewer 1 Report
Comments and Suggestions for Authors
1. The authors should address safety issues of appled chemicals, in particular Silver nitroprusside.
In raws 59-61 biocompatibility of similar films is addressed, without quoting the reference and obtained results (positive or negative ones). This should be further elaborated.
“The biocompatibility of SA- 59 AgNNPs films was also determined by toxicity assays (in vitro hemolysis and cytotoxi- 60 city)”
2. In raw 113 authors are mentioning microdillution assay and literauture 25. Further clarification is it a standard method would be usefull, so as quoting a standard as additional reference.
3. The same comment applies on assays for anti-biofilm, cytotoxicity and hemolysis assays, so as on TBC determination (raw 183). Are those methods standard, or modified ones?
4. In Fig 1, additional legend should be added on figures 1d-1h for better differentiation between 5 analysed elements.
5. In raw 256 further clarification of obtained differences between 2 samples regarding the polydispersity index is necessary. In the way presented, no discussion is given. The same applies for the discussion of zeta potential (raw 262) where the authors only mention that one sample has more negative potential. Additionall, more in depth analisys should be added.
6. The authors should quote the standard used for the determination of antimicrobial activity under the references part.
7. Quoting of the standard or short description of the method for determination of total soluble sugars in missing and should be added.
Comments on the Quality of English Language
No comments.
Author Response
Dear Editor and reviewer,
Greetings,
Thank you for your decision as the revision is associated with valuable comments on our recently submitted article. I am thankful to the Editors and reviewers for their keen observations and comments for the betterment of our article. Hence, I am submitting the revised manuscript and complete responses to the reviewers. The changes made in the revised manuscript are highlighted in the red-colored text.
Dear Reviewer
- The authors should address safety issues of appled chemicals, in particular Silver nitroprusside.
In raws 59-61 biocompatibility of similar films is addressed, without quoting the reference and obtained results (positive or negative ones). This should be further elaborated.
“The biocompatibility of SA- 59 AgNNPs films was also determined by toxicity assays (in vitro hemolysis and cytotoxi- 60 city)”
Thank you very much for your comment. We have added a description of the safety and range of applications of silver nitroprusside on lines 72-76. Further, the hemolytic properties of silver nitroprusside nanoparticles and it incorporated sodium alginate film was evaluated. In the introduction section, we highlighted experiment and their necessity. Further, the results were concluded in the results and discussion section. In the original text, lines 59-61, "The biocompatibility of the SA-AgNNPs films was also determined by toxicity tests (in vitro hemolysis and cytotoxicity). This sentence summarizes the testing of the biocompatibility of sodium nitroprusside-sodium alginate films in this thesis; the results of the hemolysis experiments are in Figures 7b and 7c, and the analysis of the results is in lines 405-418.
- In raw 113 authors are mentioning microdillution assay and literauture 25. Further clarification is it a standard method would be usefull, so as quoting a standard as additional reference.
Thank you for your comment. We have revised the manuscript to add that the experiments were performed according to the CLSI standard and cited the standard as reference 34 (Consensus statement on the adherence to Clinical and Laboratory Standards Institute (CLSI) Antimicrobial Susceptibility Testing Guidelines (CLSI-2010 and CLSI-2010-update) for Enterobacteriaceae in clinical microbiology laboratories in Taiwan. Journal of Microbiology, Immunology and Infection 2010, 43, 452-455.).
- The same comment applies on assays for anti-biofilm, cytotoxicity and hemolysis assays, so as on TBC determination (raw 183). Are those methods standard, or modified ones?
Thank you for your valuable comments. Those methods are standard but performed with little modification like sample voulme and concentration. Hence, we have mentioned the additional details in the each methods.
- In Fig 1, additional legend should be added on figures 1d-1h for better differentiation between 5 analysed elements.
Thank you for your keen observation. We have added the legend to Figure 1.
- In raw 256 further clarification of obtained differences between 2 samples regarding the polydispersity index is necessary. In the way presented, no discussion is given. The same applies for the discussion of zeta potential (raw 262) where the authors only mention that one sample has more negative potential. Additionall, more in depth analisys should be added.
Thank you for your valuable comments. We have added an analysis of the polydispersity index difference and negative potential difference between the two samples in lines 307-315 of the revised manuscript. The details are: A value of PDI <0.5 has been considered that the material was homogenous and stable [52]. A lower PDI value represents a higher size homogeneity. Therefore, it can be inferred from the PDI values that the size distribution of AgNNPs is more uniform than that of SA-AgNNPS. This can be explained by the fact that nanoparticles with a large size are prone to showing a narrow size distribution. Moreover, it can be observed from Table 1, and Figures S4, and S5, that the surface potential of AgNNPs and SA-AgNNPs are -26.97±0.21 mV and -58.40±2.36 mV, respectively. The potential of SA-AgNNPs was more negative than that of AgNNPs, due to the capping of SA on the surface of AgNNPs considering the carboxyl group in SA molecuels [53].
- The authors should quote the standard used for the determination of antimicrobial activity under the references part.
Thank you for your valuable comments. We have added a citation to the standard method for antimicrobial experiments in the revised manuscript as the reference 34 (Consensus statement on the adherence to Clinical and Laboratory Standards Institute (CLSI) Antibacterial Susceptibility Testing Guidelines (CLSI-2010 and CLSI-2010-update) for Enterobacteriaceae in clinical microbiology laboratories in Taiwan. Journal of Microbiology, Immunology, and Infection 2010, 43, 452-455.).
- Quoting of the standard or short description of the method for determination of total soluble sugars in missing and should be added.
Thank you for your valuable comments. We have added the respective reference in the revised manuscript: (Das, S.K.; Vishakha, K.; Das, S.; Ganguli, A. Antibacterial and antibiofilm activities of nanoemulsion coating prepared by using caraway oil and chitosan prolongs the shelf life and quality of bananas. Applied Food Research 2023, 3, 100300.).
Thank you very much for your valuable and insightful comments on improving our articles in a better way. We believe that we have answered all the queries asked by the reviewers and made all the changes in the manuscript as per the Editor and reviewer’s suggestions. We are looking forward to your valuable decision on our revised submission.
Thank you!
Reviewer 2 Report
Comments and Suggestions for Authors
Wang et al report new composite films synthesized via the incorporation of silver nitropruside NPs (AgNNPs) in a biobased Sodium Alginate martix that show good antimicrobial properties. This is a novel combination of nanoparticles and biobased matrices that should be published in Nanomaterials.
However, there are a few points that must be addressed before publication.
1) line 47-51. The authors should provide additional information with respect to the use of SA films for fruit preservation and the type of antimicrobials incorporated in the SA films to demonstrate the novelty of their work.
2) line 56-58: "The potential of AgNNPs in edible coatings..." This a major concern of mine. The authors should prove or at least explain how the incorporation of silver and iron nitropruside renders the film edible, particularly since the final AgNNPs loading is not provided. For example, the MSDS of sodium nitropruside suggessts high toxicity if swallowed. Similar hazards are reported for silver nitrate. I understand that the loading of AgNNPs in the SA films is very small (although not provided), but further clarification should be included.
3) line 85-88: The authors should better describe the synthesis of the films. Firstly, provide the solvent (I pressume it is water). Secondly, explain the SA concentration of 1% (is it w/w or w/v unless the solvent is water). The sentence "with a final concentration of 5 mM for both of them in 1:2 volume ratio" is very confusing, if not entirely wrong.
4) line 93-102: the authors should add more details how the FT-IR of the films was measured (was it in ATR mode?), as well as how the DLS measurements were run. I am rather confused how the particle size of the SA-AgNNPs films was measured with DLS. In fact, I am not sure what the authors mean by particle size of a film.
5) line 114-115: This is another major concern of mine. What do the authors exactly mean by the phrase "different concentration of the films", and why did they decide to do this. Moreover, I cannot fully understand how the films were prepared inside the wells. The text reads "100 mL of different concentrations of films with final concentrations of 500, .... and 7.8 μg/mL were added to...". What does "final concentration" mean and where does it refer to? Is this the nominal concentration of AgNNPs? How are the films produced? Are the plates dried in the oven as done for the films prepared on a larger scale on petri dish? I went over the SI tables, but I couldn't understand the experimetnal process. Are the new films different than the ones characterised by FTIR, TEM, SEM-EDX, etc? If this is true, then how do the authors know that the new films are of the same type with the standard ones without any characterisation?
6) line 136-7: The quantity of the AgNNPs and the films is measured in volume units (μL). I don't understand how this is possible unless a dispersion has been previously prepared. Again the phrase of "different concentrations" pops up, which is confusing.
7) Line 202-10: Firstly, the authors should provide details about the mapping colours in the figure caption. The legend on top of each image is very small and even after magnifying the page view, it is not possible to read it. Secondly, elemental analysis of light elements (C, N, O) with EDS should be done very carefully and after some type of calibration. This is why mapping shows carbon and oxygen "outside" the particles area.
8) line217: Can the authors measure the "solid state UV-vis spectrum" of the films by DRS to quantify %reflectance in the Visible as the well as in the UV range (UV-barrier properties)? If DRS is not available, then meausing the transmittance of the film by placing it inside the cuvette chamber may sometimes give acceptable results.
9) line 248 - Figure 4a XRD: Identify the reflections between 20 - 40 degrees for AgNNPs patterns. Compare the experimental pattern with the simulated one, procuded by a deposited crystal structure and provide the respective ICDD number (JCPDS card). Figure 4b FTIR: Kindly increase magnification of all spectra to clearly distinguish the peaks. Currently, the peak intensity is smaller than the marker. In addition, specify how the spectra were collected (e.g. ATR, KBR pellets etc).
10) line261-3: as mentioned above, the authors must provide details how DLS and Z-potential measurements were run on the SA-AgNNPs films. Moreover, a monodisperse population is defined by a PDI < 0.2, not a PDI > 0.1. The particle size distribution in the SI suggests a monodisperse population for AgNNPs. I presume the three figures correspond to three different measurements of the same sample? Or is three different batches?
12) lines 277-279: As mentioned above, the authors need to provide more details about the additional optimisation of the new composite films regarding the concentration of silver nitrate and sodium nitropruside.
Comments on the Quality of English Language
The level of english is good, however I must point out that the use of the verb "incorporate" in the passive voice in the title is not correct (i.e. the films are not incorporated by nanoparticles, the nanoparticles are incorporated in the films).
Author Response
Dear Editor and reviewer,
Greetings,
Thank you for your decision as the revision is associated with valuable comments on our recently submitted article. I am thankful to the Editors and reviewers for their keen observations and comments for the betterment of our article. Hence, I am submitting the revised manuscript and complete responses to the reviewers. The changes made in the revised manuscript are highlighted in the red-colored text.
Dear Reviewer, 3,
Wang et al report new composite films synthesized via the incorporation of silver nitropruside NPs (AgNNPs) in a biobased Sodium Alginate martix that show good antimicrobial properties. This is a novel combination of nanoparticles and biobased matrices that should be published in Nanomaterials.
However, there are a few points that must be addressed before publication.
1) line 47-51. The authors should provide additional information with respect to the use of SA films for fruit preservation and the type of antimicrobials incorporated in the SA films to demonstrate the novelty of their work.
Thank you for your comments. We have added information about the specific type of antimicrobial added to the SA in line 65-66 of the revised manuscript, specifically: To further enhance the performance of SA coatings, additives, especially antibacterial agents such as plant extracts [23], essential oils [24], phenolics [25], and nanoparticles [26] are often incorporated into the coatings to prolong the shelf life of food products.
2) line 56-58: "The potential of AgNNPs in edible coatings..." This a major concern of mine. The authors should prove or at least explain how the incorporation of silver and iron nitropruside renders the film edible, particularly since the final AgNNPs loading is not provided. For example, the MSDS of sodium nitropruside suggessts high toxicity if swallowed. Similar hazards are reported for silver nitrate. I understand that the loading of AgNNPs in the SA films is very small (although not provided), but further clarification should be included.
Thank you for your question. We have modified the inappropriate statement about edible coatings by replacing edible coatings with food coatings.
We have added the final loading of AgNNPs in line 110-111 in the revised manuscript, specifically: The final 1 mL SA-AgNNPs film-forming solution contained about 1.28 mg of AgNNPs.
We have added a description of the safety and range of applications for sodium nitroprusside on line 68-71.
3) line 85-88: The authors should better describe the synthesis of the films. Firstly, provide the solvent (I pressume it is water). Secondly, explain the SA concentration of 1% (is it w/w or w/v unless the solvent is water). The sentence "with a final concentration of 5 mM for both of them in 1:2 volume ratio" is very confusing, if not entirely wrong.
Thank you for your comments. We have added 1% as w/v in line 108 of the revised manuscript, and also revised the description of the thin film synthesis process, specifically: 90 mL of 1% (w/v) aqueous SA solution was configured and 25.48 mg of AgNO3 and 89.39 mg of SNP were added to the aqueous SA solution. It was stirred vigorously for 18 h at room temperature and in dark conditions.
4) line 93-102: the authors should add more details how the FT-IR of the films was measured (was it in ATR mode?), as well as how the DLS measurements were run. I am rather confused how the particle size of the SA-AgNNPs films was measured with DLS. In fact, I am not sure what the authors mean by particle size of a film.
Thank you for your comments. We have added explanation in the revised draft that the FTIR was measured in ATR mode, specifically: The functional properties of AgNNPs, SA films, and SA-AgNNPs films were characterized through attenuated total reflectance-Fourier transform infrared (ATR-FTIR) spectroscopy (FTIR PerkinElmer Paragon 500, USA).
Regarding the experimental method for particle size analysis of thin films, we tested the particle size, dispersion, and ζ-potential of AgNNPs in the pre-film-forming solution of SA-AgNNPs. We have modified the test method for the determination of Zeta potentials, size, and particle distribution of AgNNPs in the pre-film-forming solution in line 125-129 of the revised manuscript as follows: To determine the particle size, dispersion, and ζ-potential of AgNNPs and SA-AgNNPs films, disperse 200 µg of AgNNPs in PBS (1 mL) and 1 mL of SA-AgNNPs pre-film-forming solution in PBS (5 mL) and sonicate for 30 seconds. Then analyzed by dynamic light scattering (DLS) and electrolytic light scattering (ELS) analysis (Malvern PANalytical Netherland).
5) line 114-115: This is another major concern of mine. What do the authors exactly mean by the phrase "different concentration of the films", and why did they decide to do this. Moreover, I cannot fully understand how the films were prepared inside the wells. The text reads "100 mL of different concentrations of films with final concentrations of 500, .... and 7.8 μg/mL were added to...". What does "final concentration" mean and where does it refer to? Is this the nominal concentration of AgNNPs? How are the films produced? Are the plates dried in the oven as done for the films prepared on a larger scale on petri dish? I went over the SI tables, but I couldn't understand the experimetnal process. Are the new films different than the ones characterised by FTIR, TEM, SEM-EDX, etc? If this is true, then how do the authors know that the new films are of the same type with the standard ones without any characterisation?
Thank you for your constructive comments. In the manuscript, the different concentrations of film refer to the concentration (w/v) of the film when it is dissolved in water, and serial dilution with water to obtain different concentrations (w/v) of film solution. The final concentration is, for example, if an aqueous film solution of the original concentration of 1000 μg/mL is mixed with an equal volume of culture medium containing bacteria in a test, the aqueous film solution is diluted by the culture medium by a factor of two, resulting in a final concentration of 500 μg/mL. I have made a change in line 145-147 of the revised manuscript, specifically: The SA-AgNNPs films were dissolved in distilled water and continued to be diluted continuously with distilled water to finally obtain aqueous solutions of SA-AgNNPs films at different concentrations (w/v), and line 168-169: AgNNPs, SA films were diluted in the same way as SA-AgNNPs films in subsection 2.6.
6) line 136-7: The quantity of the AgNNPs and the films is measured in volume units (μL). I don't understand how this is possible unless a dispersion has been previously prepared. Again the phrase of "different concentrations" pops up, which is confusing.
Thank you for your question. In the revised manuscript, the different concentrations of films are films dissolved in water (w/v), and the aqueous solutions of films with different mass concentrations were prepared in the experiments, and 10 μL of the aqueous solutions of films with different mass concentrations were added to each well during the cytotoxicity test. I have added in line 165-166 of the revised manuscript, specifically: AgNNPs, SA films were diluted in the same way as SA-AgNNPs films in subsection 2.6.
7) Line 202-10: Firstly, the authors should provide details about the mapping colours in the figure caption. The legend on top of each image is very small and even after magnifying the page view, it is not possible to read it. Secondly, elemental analysis of light elements (C, N, O) with EDS should be done very carefully and after some type of calibration. This is why mapping shows carbon and oxygen "outside" the particles area.
Thank you for your question. We have added details about mapping colours to the illustration in Figure 1 of the revised manuscript, specifically: (yellow is elemental silver, red is elemental iron, green is elemental carbon, purple is elemental nitrogen, and blue is elemental oxygen.), and an enlarged font legend has been added to Figure I. Further, we noted that the C and O was exist outside of the particles which due to the properties of grid as well as oxidation under the electron beam. However, further reasons are unclear.
8) line217: Can the authors measure the "solid state UV-vis spectrum" of the films by DRS to quantify %reflectance in the Visible as the well as in the UV range (UV-barrier properties)? If DRS is not available, then meausing the transmittance of the film by placing it inside the cuvette chamber may sometimes give acceptable results.
Thank you for your question. We have added the data on film transmittance in Figure 3b of the revised manuscript and the method of determining film transmittance in line 119-121 of the revised manuscript. We have added an analysis of the film transmittance in line 259-268 of the revised version, specifically: In transparent food packaging, light reduces food quality and accelerates spoilage. Therefore, food packaging with light-blocking protects the food [10]. The UV-vis spectrophotometric light blocking ability of the films is shown in Figure 3b. The transmittance of the control SA film was 0.69% at 200 nm while the maximum transmittance at 700 nm was 77.75%. The transmittance at the same wavelength decreased sharply to 0.01% and 54.43%, respectively, with the addition of AgNNPs compared to the control SA film. The films containing AgNNPs exhibited significant barrier properties to UV and visible light, it was attributed to the nanoparticles dispersed in the film matrix impeded the passage of light or scattered the light [44]. According to our results, SA films containing AgNNPs can be used for UV protection packaging.
9) line 248 - Figure 4a XRD: Identify the reflections between 20 - 40 degrees for AgNNPs patterns. Compare the experimental pattern with the simulated one, procuded by a deposited crystal structure and provide the respective ICDD number (JCPDS card). Figure 4b FTIR: Kindly increase magnification of all spectra to clearly distinguish the peaks. Currently, the peak intensity is smaller than the marker. In addition, specify how the spectra were collected (e.g. ATR, KBR pellets etc).
Thank you for your question. We have added the JCPDS card information for AgNNPs in line 277 of the revised manuscript. We have also added the X-ray diffraction curves of Simulated-AgNNPs in Fig. 4a. And we have added the analysis in line 276-278 of the revised manuscript, specifically: consistent with the displayed peak positions of simulated-AgNNPs, showed the successful synthesis of AgNNP (JCPDS No.: 052-0366) [45]. And in the revision of Fig. 4b to make the peaks look more pronounced in the figure. And added in line 120 in the revised manuscript that the FTIR was measured in ATR mode, specifically: The functional properties of AgNNPs, SA films, and SA-AgNNPs films were characterized through attenuated total reflectance-Fourier transform infrared (ATR-FTIR) spectroscopy (FTIR PerkinElmer Paragon 500, USA).
10) line261-3: as mentioned above, the authors must provide details how DLS and Z-potential measurements were run on the SA-AgNNPs films. Moreover, a monodisperse population is defined by a PDI < 0.2, not a PDI > 0.1. The particle size distribution in the SI suggests a monodisperse population for AgNNPs. I presume the three figures correspond to three different measurements of the same sample? Or is three different batches?
Thank you for your comments. We have corrected the sentence in the characterization of this section in line 307-308 of the revised version, specifically: A value of PDI<0.5 has been considered that the material was homogenous and stable [52]. The three SI plots correspond to three different measurements of the same sample.
12) lines 277-279: As mentioned above, the authors need to provide more details about the additional optimisation of the new composite films regarding the concentration of silver nitrate and sodium nitropruside.
Thank you for your valuable comments. We have changed and added line 326-334 in the revised manuscript, specifically: First, the SA concentration and the concentration ratio of AgNO3 to SNP used to synthesize the AgNNPs in the films were optimized to obtain the best antibacterial properties for further experiments. As shown in Fig. S6-10 and Table S1. Specifically, AgNO3 and SNP were added to 0.25%, 0.5%, 1%, and 2% (w/v) SA aqueous solutions using concentration ratios of 1:1, 1:2, and 2:1, respectively. The antibacterial properties of the resulting samples were examined. According to Table S1, the 1% (w/v) aqueous solution of SA was added at a concentration ratio of 1:2 between AgNO3 and SNP was the optimal synthesis ratio for the antibacterial properties. Therefore, this ratio was chosen for subsequent experiments.
Comments on the Quality of English Language
The level of english is good, however I must point out that the use of the verb "incorporate" in the passive voice in the title is not correct (i.e. the films are not incorporated by nanoparticles, the nanoparticles are incorporated in the films).
Thank you for your valuable comments. We have changed the title, specifically: “Development and Analysis of silver nitroprusside nanoparticles Incorporated Sodium Alginate Films for Banana Browning Prevention.”
Thank you very much for your valuable and insightful comments on improving our articles in a better way. We believe that we have answered all the queries asked by the reviewers and made all the changes in the manuscript as per the Editor and reviewer’s suggestions. We are looking forward to your valuable decision on our revised submission.
Thank you!
Reviewer 3 Report
Comments and Suggestions for Authors
Reviewer comments and suggestions
The authors should include the page and line numbers of the revised text in the next revision round for the reviewers to track changes. The revisions that you make should be visible in both the review report and in the revised manuscript. Please pay attention to all the comments and suggestions very carefully from the reviewer and address each one by one.
Abstract (Page 1)
(1) Ag2[Fe(CN)5NO] is not interpreted, Please do that.
(2) Please include numerical values (statistical values) to support your conclusion.
Introduction (Page 1& 2)
(1) The flow of the introduction is acceptable (good). But the introduction should contain previous/ recent work carried out by other researchers, Please cite them properly.
(2) Line 45- Authors mentioned the properties of SA. Why did you choose SA from the other available natural materials? How does SA differ or better from other natural materials which have been used for food coating? It can be explained comparatively. Please revise.
(3) Please avoid complex sentences. Try to include short but meaningful sentences.
Methodology
(1) All the chemical formulas should be interpreted in the manuscript. Line 79? Please revise. This comment applies to the whole document.
(2) Line 71-73- Please include only the ATCC numbers within brackets. No need to repeat the organisms’ names.
PLease Revise.
(3) Line 86- 1%? Is it w/v?
Please Mention and Revise.
(4) Line 88- Avoid using terms such as ‘after that, then, finally and next’ in scientific writing.
Please Revise.
(5) Line 91-92- Is this your control? Not clear.
PLease Mention that.
(6) Line 93- It should be characterization of….
Please Revise.
(7) Please avoid complex sentences. Try to include short but meaningful sentences.
(8) Line 103- Authors have not mentioned the preparation of the bacterial cultures with suitable cell concentrations for well diffusion assay. Please Revise.
(9) Line 110-120- Avoid complex sentences. Preparation of bacterial cultures before the MIC and MBC is missing in the text. Please Revise.
Results
(1) Overall results section is presented well with appropriate findings and figures.
(2) The results need to be presented in the past tense (Line 229, 263, 281,380 etc.). Revise. This comment applies to the whole results section.
(3) There are three images in each figure S5-9. Label them appropriately. (Line 279)
(4) Line 282- How do you say ‘SA-AgNNPs films exhibit excellent antibacterial activities’? compared to what? Did you statistically analyse this? Or else please avoid using such vague statements. Revise.
(5) You should interpret TCH where it appears the first time in the text. Line 287
(6) There is no mention of control for the well-diffusion test. What is the control antimicrobial agent? Considering the pathogenic bacteria chosen in this study, it’s usually difficult to choose just one control for all five bacteria. Justification required.
(7) Figure 6- Why did you measure the OD up to 24 h?
(8) Line 436- ‘no significant difference’? if you use this term, it should be related with statistical p values. Revise.
(9) Changes in total bacterial colonies (TBC)- this part is a little unclear. Were you able to count the colonies clearly and separately? The colonies are not visible in figures.
(10) Line 446-447- How do you say this is excellent? Compared to what? Avoid these vague words. Also, avoid repeating words like excellent.
(11) Figure 8- What is meant by CK? Revise in the manuscript.
(12) Line 432-448- Sentences are too long. Difficult to grasp the data. Please revise long sentences throughout the manuscript.
(13) What is the importance of finding total soluble sugar content in the skin and pulp? Why is this relevant? Discuss in the text.
(14) Overall, results and discussion section is well written. However there is a lack of proper citations of previous similar findings. Please support the data with more citations.
Conclusion
(1) Conclusion is incomplete. It needs to be supported with the values obtained. Please Revise
(2) No indication of future work is a limitation. Please add future work.
Author Response
Dear Editor and reviewer,
Greetings,
Thank you for your decision as the revision is associated with valuable comments on our recently submitted article. I am thankful to the Editors and reviewers for their keen observations and comments for the betterment of our article. Hence, I am submitting the revised manuscript and complete responses to the reviewers. The changes made in the revised manuscript are highlighted in the red-colored text.
Dear Reviewer,
Reviewer comments and suggestions
The authors should include the page and line numbers of the revised text in the next revision round for the reviewers to track changes. The revisions that you make should be visible in both the review report and in the revised manuscript. Please pay attention to all the comments and suggestions very carefully from the reviewer and address each one by one.
Abstract (Page 1)
(1) Ag2[Fe(CN)5NO] is not interpreted, Please do that.
Thank you for your question. We have modified Ag2[Fe(CN)5NO] to silver nitroprusside nanoparticles of the revised manuscript .
(2) Please include numerical values (statistical values) to support your conclusion.
Thank you for your question. We have added some specific values of the results in the summary of the revised manuscript (in lines 14-15, 22-23).
Introduction (Page 1& 2)
(1) The flow of the introduction is acceptable (good). But the introduction should contain previous/ recent work carried out by other researchers, please cite them properly.
Thank you for your question. We have added lines 56-61 to the original article with research from other researchers on the use of sodium alginate for food preservation, as well as lines 76-77 with additional citations to research from other researchers on silver nitroprusside nanoparticles.
(2) Line 45- Authors mentioned the properties of SA. Why did you choose SA from the other available natural materials? How does SA differ or better from other natural materials which have been used for food coating? It can be explained comparatively. Please revise.
Thank you for your question. We have added the advantages of sodium alginate over other natural compounds in lines 52-56.
(3) Please avoid complex sentences. Try to include short but meaningful sentences.
Thank you for your question. I have checked and changed the long sentences in the revised manuscript, for example, the long sentences in lines 440 through 441 have been changed to: Therefore, weight loss is recognized as one of the most critical parameters for predicting postharvest fruit deterioration.
Methodology
(1) All the chemical formulas should be interpreted in the manuscript. Line 79? Please revise. This comment applies to the whole document.
Thank you for your question. We have added the full names of silver nitrate and sodium nitroprusside in lines 101-102 of revised manuscript.
(2) Line 71-73- Please include only the ATCC numbers within brackets. No need to repeat the organisms’ names.
Thanks to your suggestion. We have added the respective ATCC number for each bacterium and corrected the organisms name accordingly.
PLease Revise.
(3) Line 86- 1%? Is it w/v?
Thanks for your comments. 1% is w/v. We have added the related description (w/v) in line 108 in the revised manuscript.
Please Mention and Revise.
(4) Line 88- Avoid using terms such as ‘after that, then, finally and next’ in scientific writing.
Please Revise.
Thank you for your question. We have checked and removed these phrases in the revised manuscript.
(5) Line 91-92- Is this your control? Not clear.
PLease Mention that.
Thank you for your question. We have changed the sentence in line 113 to read as follows: Pure SA film was used as a control.
(6) Line 93- It should be characterization of….
Thank you for your suggestion. We have corrected in the revised manuscript.
Please Revise.
(7) Please avoid complex sentences. Try to include short but meaningful sentences.
Thank you for your suggestion. I have checked and changed the long sentences in the revised manuscript, for example, the long sentences in lines 440 through 441 have been changed to: Therefore, weight loss is recognized as one of the most critical parameters for predicting postharvest fruit deterioration.
(8) Line 103- Authors have not mentioned the preparation of the bacterial cultures with suitable cell concentrations for well diffusion assay. Please Revise.
Thank you for your comment. We have added instructions on how to prepare bacterial cultures on lines 133-134. Specifically: for bacterial inoculation, a sterile cotton swab was dipped in bacterial solution (107 CFU/mL) and evenly cultured on the Muller-Hinton agar (MHA) plates.
(9) Line 110-120- Avoid complex sentences. Preparation of bacterial cultures before the MIC and MBC is missing in the text. Please Revise.
Thank you for your question. We have shortened the long sentence in this section, and I have added instructions on how to prepare bacterial cultures of the of revised manuscript. The details are: Briefly, the pathogens were initially cultured in Mueller-Hinton Broth (MHB) for 24 h.
Results
(1) Overall results section is presented well with appropriate findings and figures.
Thank you for your positive comments.
(2) The results need to be presented in the past tense (Line 229, 263, 281,380 etc.). Revise. This comment applies to the whole results section.
Thank you for your comment. We have checked and revised the tenses in both the results and discussion sections.
(3) There are three images in each figure S5-9. Label them appropriately. (Line 279)
Thank you for your comment. We have labeled Figure S6-10 in the supplementary file with a, b to differentiate between the three figures and revised the figure notes for Figure S6-10.
(4) Line 282- How do you say ‘SA-AgNNPs films exhibit excellent antibacterial activities’? compared to what? Did you statistically analyse this? Or else please avoid using such vague statements. Revise.
Thank you for your comment. We have made a change in lines 336-337 of revised manuscript, the change is as follows: Unsurprisingly, the SA-AgNNPs films showed significant antimicrobial activity against all tested bacterial pathogens compared to the pure SA films (P<0.05).
(5) You should interpret TCH where it appears the first time in the text. Line 287
Thank you for your sugestion. We have added the full name of TCH in line 139.
(6) There is no mention of control for the well-diffusion test. What is the control antimicrobial agent? Considering the pathogenic bacteria chosen in this study, it’s usually difficult to choose just one control for all five bacteria. Justification required.
Thank you for your suggestion. We have added TCH as a positive control in line 139, and since TCH is a broad-spectrum antimicrobial agent, it was chosen for this paper as a positive control for experiments against different bacteria.
(7) Figure 6- Why did you measure the OD up to 24 h?
Bacterial growth curves are measured over 24 hours for the following reasons:
- Experimental feasibility: Under laboratory conditions, culturing bacteria and determining their growth is a time-consuming process. The inoculation, culture and growth curve of bacteria can be done within 24 hours, which is in line with the efficiency of the experimental operation.
- Bacterial growth characteristics: In the early stage of bacterial growth, the number of cells grows rapidly and the number of colonies and biomass increase significantly. This stage usually reaches the logarithmic phase of bacterial growth within 24 hours, i.e. the period of fastest bacterial growth. The physiological characteristics and metabolic activities of the bacteria in the logarithmic phase are also relatively stable, which is more conducive to the study of bacterial growth pattern and determination of relevant parameters.
- Long incubation time will lead to aging of the bacteria and changes in cell morphology and physiological characteristics, which will affect the accuracy of the experimental results. Selecting the incubation time within 24 hours can avoid this problem.
Therefore, the choice of determining the bacterial growth curve within 24 hours is to ensure the feasibility of the experiment, the stability of the bacterial growth characteristics, as well as the accuracy and reliability of the experimental results.
(8) Line 436- ‘no significant difference’? if you use this term, it should be related with statistical p values. Revise.
Thanks for your suggestions. The results were presented statistically.
(9) Changes in total bacterial colonies (TBC)- this part is a little unclear. Were you able to count the colonies clearly and separately? The colonies are not visible in figures.
Thank you for your comments. The experimental method We used in the assay for TBC was to dilute the supernatant of banana pulp and peel at different multiplications, incubate it overnight on MHA medium, choose a dilution plate with an average number of colonies ranging from 30 to 300 for counting, and then multiply the dilutions together for the result.
(10) Line 446-447- How do you say this is excellent? Compared to what? Avoid these vague words. Also, avoid repeating words like excellent.
Thank you for your suggestions. We have checked and corrected them in the revised manuscript.
(11) Figure 8- What is meant by CK? Revise in the manuscript.
Thank you for your comment. We have changed CK to uncoated group in the figure note of Figure 8.
(12) Line 432-448- Sentences are too long. Difficult to grasp the data. Please revise long sentences throughout the manuscript.
Thanks for the question. We have shortened the long sentence in lines 488-504 and checked and revised the long sentence in the revised manuscript.
(13) What is the importance of finding total soluble sugar content in the skin and pulp? Why is this relevant? Discuss in the text.
Thank you for your question. We have added a note on the relationship between total soluble sugars and post-harvest indicators for bananas in lines 461-462.
(14) Overall, results and discussion section is well written. However, there is a lack of proper citations of previous similar findings. Please support the data with more citations.
Thanks to your question. We added references to similar findings, for example: references 54 (Antimicrobial activity of metal oxide nanoparticles against Gram-positive and Gram-negative bacteria: a comparative study. International journal of nanomedicine 2012, 6003-6009.), 58 (Antibacterial effects of silver nanoparticles stabilized in solution by sodium alginate. Biochemistry & molecular biology journal 2016, 2.), 66 (Enhanced functional properties of chitosan films incorporated with curcumin-loaded hollow graphitic carbon nitride nanoparticles for bananas preservation. Food Chemistry 2022, 366, 130539.).
Conclusion
(1) Conclusion is incomplete. It needs to be supported with the values obtained. Please Revise
Thanks for the question. We added specific data in lines 512-518, 522-527.
(2) No indication of future work is a limitation. Please add future work.
Thank you for your question. We have added lines 530-533 about future work.
Thank you very much for your valuable and insightful comments on improving our articles in a better way. We believe that we have answered all the queries asked by the reviewers and made all the changes in the manuscript as per the Editor and reviewer’s suggestions. We are looking forward to your valuable decision on our revised submission.
Thank you!
Round 2
Reviewer 2 Report
Comments and Suggestions for Authors
The authors have adequately addressed all my comments and suggestions. In my opinion, the manuscript is in far better shape and can be published in Nanomaterials after some minor revisions.
1) I believe PBS (phosphate buffer solution) is only included as an abbreviation (i.e the full name has been omitted).
2) The same applies for EDX. The full name must be provided. In addition, the type of EDX detector coupled to the FE-SEM must be included in the materials and methods section (paragraph 2.4).
2) line 246-7: Rephrase the caption as: (a) Transmission electron microscopy analysis of AgNNPs with elemental mappings: silver (yellow), iron (red) etc.... (b) EDX spectrum (not elemental spectrum)
3) lines 261-270: I am glad to find out that the authors did measure the optical properties of the films. Apparently, these show decent UV-barrier properties. This is good. To clearly demonstrate their case, the authors should not focus on the actual transmittance value at 200 nm, which is practically zero for both control and AgNNPs-SA films, but point out instead that the synthesized films have a UV cut-off close to 280 nm (adding a vertical dashed line on the figure should help). This is the key point to demonstrate UV-barrier properties, i.e that no light passes through the film at a UV wavelenth below xxx nm (ideally this should be up to 400 nm to cover the whole UV range).
4) line 297: The revised XRD figure and following section is now good. However, a better "simulated pattern" of AgNNPs must be provided in Figure 4a.
If these minor revisions are addressed, then the paper can be published in Nanomaterials without any further review.
Comments on the Quality of English Language
The new title is good, but I believe its "incorporated in". If the title is too long, then "analysis" can go (development directly implies analysis).
Author Response
Dear Editor and Reviewer:
Greetings,
Thank you for your decision, as the revision is associated with valuable comments on our recently submitted article. I am thankful to the Editors and reviewers for their keen observations and comments for the betterment of our article. Hence, I am submitting the revised manuscript and complete responses to the reviewers. The changes made in the revised manuscript are highlighted in the green-colored text.
Reviewer -2
The authors have adequately addressed all my comments and suggestions. In my opinion, the manuscript is in far better shape and can be published in Nanomaterials after some minor revisions.
1) I believe PBS (phosphate buffer solution) is only included as an abbreviation (i.e the full name has been omitted).
Response: Thank you for your keen observation. We have added the full name of PBS on the revised manuscript.
2) The same applies for EDX. The full name must be provided. In addition, the type of EDX detector coupled to the FE-SEM must be included in the materials and methods section (paragraph 2.4).
Response: We have added the full name of EDX on the revised manuscript, specifically: energy-dispersive X-ray spectroscopy. The type of EDX detector has been added, specifically: (JEM-2100F, JEOL, Japan).
2) line 246-7: Rephrase the caption as: (a) Transmission electron microscopy analysis of AgNNPs with elemental mappings: silver (yellow), iron (red) etc.... (b) EDX spectrum (not elemental spectrum)
Response: We have revised this section in the revised manuscript, specifically: Figure 1. Transmission electron microscopy analysis of AgNNPs with elemental mappings: silver (yellow), iron (red), carbon (green), nitrogen (purple), oxygen (blue) (a) and EDX spectrum (b).
3) lines 261-270: I am glad to find out that the authors did measure the optical properties of the films. Apparently, these show decent UV-barrier properties. This is good. To clearly demonstrate their case, the authors should not focus on the actual transmittance value at 200 nm, which is practically zero for both control and AgNNPs-SA films, but point out instead that the synthesized films have a UV cut-off close to 280 nm (adding a vertical dashed line on the figure should help). This is the key point to demonstrate UV-barrier properties, i.e that no light passes through the film at a UV wavelenth below xxx nm (ideally this should be up to 400 nm to cover the whole UV range).
Response: Thank you so much for your positive response and guidance. We have added vertical dashed lines to Figure 3b of the revised manuscript and emphasized the difference in the transmittance of the two films at 280 nm.
4) line 297: The revised XRD figure and following section is now good. However, a better "simulated pattern" of AgNNPs must be provided in Figure 4a.
Response: We have modified the "simulated pattern" of AgNNPs in the revised manuscript in Figure 4a. We have provided new simulation data and plotted it using a vertical line approach.
If these minor revisions are addressed, then the paper can be published in Nanomaterials without any further review.
Thank you very much for your valuable and insightful comments on improving our articles in a better way. We believe that we have answered all the queries asked by the reviewers and made all the changes in the manuscript as per the Editor and reviewer’s suggestions. We are looking forward to your valuable decision on our revised submission.
Thank you!
Reviewer 3 Report
Comments and Suggestions for Authors
Authors have improved manuscript.
some style modifications are required.
Line 324
3.2. Antibacterial activity of SA-AgNNPs films
please include 12 pt space before the line and 3 pt after the line for the title of each subsection
line 325
3.2.1. Well diffusion assay of SA-AgNNPs films
please include 12 pt space before the line and 3 pt after the line for the title of each subsubsection
Line 371
3.2.3. Anti-biofilm assay of SA-AgNNPs films
Same comment as previous
Line 392 and line 393
3.3. Security assessment AgNNPs, SA films, and SA-AgNNPs films
3.3.1. Cytotoxicity assays of SA-AgNNPs films 393
same comment as previous
Table 2
Please indicate what are a,b,c,d below the table
Author Response
Dear Editor and reviewer,
Greetings,
Thank you for your decision as the revision is associated with valuable comments on our recently submitted article. I am thankful to the Editors and reviewers for their keen observations and comments for the betterment of our article. Hence, I am submitting the revised manuscript and complete responses to the reviewers. The changes made in the revised manuscript are highlighted in the blue-colored text.
Reviewer 3
Authors have improved manuscript.
some style modifications are required.
Line 324
3.2. Antibacterial activity of SA-AgNNPs films
please include 12 pt space before the line and 3 pt after the line for the title of each subsection
Response: We have revised the title of section 3.2 to 12 pt space before the line and 3 pt after the line.
line 325
3.2.1. Well diffusion assay of SA-AgNNPs films
please include 12 pt space before the line and 3 pt after the line for the title of each subsubsection
Response: We have revised the title of section 3.2.1 to 12 pt space before the line and 3 pt after the line.
Line 371
3.2.3. Anti-biofilm assay of SA-AgNNPs films
Same comment as previous
Response: We have revised the title of section 3.2.3 to 12 pt space before the line and 3 pt after the line.
Line 392 and line 393
3.3. Security assessment AgNNPs, SA films, and SA-AgNNPs films
3.3.1. Cytotoxicity assays of SA-AgNNPs films 393
same comment as previous
Response: We have revised the title of section 3.3 and 3.3.1 to 12 pt space before the line and 3 pt after the line.
Table 2
Please indicate what are a,b,c,d below the table
Response: We have indicated what are a,b,c, and d below Table 2 in the revised manuscript.
Thank you very much for your valuable and insightful comments on improving our articles in a better way. We believe that we have answered all the queries asked by the reviewers and made all the changes in the manuscript as per the Editor and reviewer’s suggestions. We are looking forward to your valuable decision on our revised submission.
Thank you!